# Advancing Breast Cancer Heterogeneity Analysis: Insights from Genomics, Transcriptomics and Proteomics at Bulk and Single-Cell Levels

**DOI:** 10.3390/cancers15164164

**Published:** 2023-08-18

**Authors:** Zijian Zhu, Lai Jiang, Xianting Ding

**Affiliations:** 1State Key Laboratory of Oncogenes and Related Genes, Institute for Personalized Medicine, Shanghai Jiao Tong University, Shanghai 200030, China; zzj2000@sjtu.edu.cn; 2Department of Anesthesiology and Surgical Intensive Care Unit, Xinhua Hospital, School of Medicine and School of Biomedical Engineering, Shanghai Jiao Tong University, Shanghai 200025, China; jianglai@xinhuamed.com.cn

**Keywords:** breast cancer, heterogeneity, single-cell genome, single-cell transcriptome, single-cell proteome

## Abstract

**Simple Summary:**

This review discusses the heterogeneity of breast cancer and strategies to understand it using multi-omics profiling methods at the traditional bulk and single-cell levels. It begins with the exploration of genomics and its role in classifying breast cancer into molecular subtypes. Following that, it delves into the contribution of transcriptomics in illuminating the diverse gene expression patterns within individual breast cancer cells, further underscoring the disease’s complexity. The narrative then addresses the impact of proteomics, which unveils intricate protein expression patterns and modifications. Despite these advancements, it is essential to integrate multiple omics approaches to fully unravel breast cancer heterogeneity. The integration of genomics, transcriptomics, and proteomics at the single-cell level holds great promise for advancing breast cancer research and management in the future.

**Abstract:**

Breast cancer continues to pose a significant healthcare challenge worldwide for its inherent molecular heterogeneity. This review offers an in-depth assessment of the molecular profiling undertaken to understand this heterogeneity, focusing on multi-omics strategies applied both in traditional bulk and single-cell levels. Genomic investigations have profoundly informed our comprehension of breast cancer, enabling its categorization into six intrinsic molecular subtypes. Beyond genomics, transcriptomics has rendered deeper insights into the gene expression landscape of breast cancer cells. It has also facilitated the formulation of more precise predictive and prognostic models, thereby enriching the field of personalized medicine in breast cancer. The comparison between traditional and single-cell transcriptomics has identified unique gene expression patterns and facilitated the understanding of cell-to-cell variability. Proteomics provides further insights into breast cancer subtypes by illuminating intricate protein expression patterns and their post-translational modifications. The adoption of single-cell proteomics has been instrumental in this regard, revealing the complex dynamics of protein regulation and interaction. Despite these advancements, this review underscores the need for a holistic integration of multiple ‘omics’ strategies to fully decipher breast cancer heterogeneity. Such integration not only ensures a comprehensive understanding of breast cancer’s molecular complexities, but also promotes the development of personalized treatment strategies.

## 1. Introduction

Breast cancer is one of the most prevalent cancers in women globally. In 2012, there were 464,000 diagnosed cases of breast cancer and 131,000 deaths among European women [1]. In 2020, breast cancer accounted for 2.26 million new cases globally, surpassing lung cancer (2.2 million cases) to become the most commonly diagnosed cancer worldwide (Figure 1). In the same year, the disease caused an estimated 685,000 deaths [1,2,3]. Projections suggest that by 2030, new cases will reach 3.9 million globally, with fatalities rising to 766,000 [4].

Despite considerable strides in both laboratory research and the clinical practice of breast cancer, the global incidence and mortality rates continue to rise. At the root of this persisting issue is the heterogeneous nature of breast cancer, which is not a monolithic disease, but a spectrum of distinct subtypes, each representing a unique malignancy within the breast’s cellular makeup. Current research categorizes these molecular subtypes into six major classes: (i) hormone-receptor-positive breast cancer (ER+), (ii) hormone receptor/HER2-positive breast cancer (ER+/HER2+), (iii) HER2-positive breast cancer (HER2+), (iv) basal-like breast cancer, (v) claudin-low subtype, and (vi) normal-like subtype [5]. The distinct subtypes of breast cancer each present unique clinical features and associated risk factors. This diversity extends to treatment responses and long-term patient survival, which differ significantly across the subtypes. This inherent complexity adds layers of challenge to the effective diagnosis and treatment of breast cancer. For instance, the basal-like subtype of breast cancer, characterized by high rates of cellular proliferation, is associated with distinct risk factors, which include the early onset of menstruation, a younger age at the first full-term pregnancy, and the accumulation of abdominal fat. In contrast, patients with the claudin-low subtype, marked by an enrichment of epithelial–mesenchymal transition markers, typically exhibit pronounced invasiveness. Such individuals often bear the burden of exposure to chemicals and radiation in their early years, leading to a high load of DNA damage induced by cancer genes and early chromosomal instability (CIN) [6].

The inherent heterogeneity of breast cancer presents considerable hurdles for conventional diagnostic and therapeutic approaches. Generally, traditional methods depend on analyzing bulk tumor tissue samples, a process which, by considering the average expression levels, may obscure the underlying heterogeneity, complicating accurate tumor classification. But emerging technologies such as single-cell analysis techniques offer promising alternatives, already being widely used in oncology research. These techniques, by investigating gene expression, phenotypes, protein levels, and other cellular properties at an individual cell level, are well suited to address the challenge of tumor heterogeneity (Figure 2) [7,8,9]. Particularly for highly heterogeneous cancers like breast cancer, a single-cell analysis can help to predict cellular evolution during tumor progression. The analysis of genetic and epigenetic variations as well as gene expression at the single-cell level is among the techniques that enhance the precision of predicting tumor development trends, evaluating treatment outcomes, and forecasting patient prognosis. Furthermore, single-cell analysis techniques play a crucial role in devising novel therapeutic strategies. These methods allow for the examination of genetic variations and phenotypic characteristics of tumor cells in detail, which can lead to the identification of new therapeutic targets. Consequently, this paves the way for the development of highly targeted, precise treatment strategies, enhancing the ability to predict treatment efficacy and potential drug resistance.

Single-cell gene sequencing, leveraging the power of next-generation sequencing (NGS), has emerged as a pivotal tool for investigating breast cancer heterogeneity [10]. Unlike traditional Sanger sequencing, NGS systems use massive parallel sequencing to yield billions of DNA reads, from 36 to 150 base pairs, which can be aligned to the human genome. This alignment allows for the detection of various genetic variations, including single-nucleotide mutations, small insertions/deletions, and copy number variations, offering a comprehensive view for streamlining the development of targeted treatment strategies. A case in point is HER2-positive breast cancer, where single-cell sequencing detects the diversity in HER2 gene amplification across different cells, facilitating the formulation of bespoke treatment plans [11]. The versatile utility of NGS extends to RNA sequencing (RNA-seq), which is a high-throughput technique enabling quantitative and sequence analyses of diverse RNA types, along with their expression levels in cells and tissues. RNA-seq facilitates an in-depth exploration of gene expression regulation, signaling, and metabolic pathways pertinent to breast cancer, thereby enriching our understanding of its molecular mechanisms. Intriguingly, beyond the recognized impact of non-coding RNAs on breast cancer progression, even the half-life of mRNA serves as an informative marker [12].

In molecular profiling, the strength of the correlation between molecular patterns and cellular behavior is pivotal, as a higher correlation implies a more accurate reflection of the tumor’s actual condition [13]. Complementing single-cell genomics and transcriptome, the emergence of single-cell proteomics offers another potent instrument for investigating breast cancer heterogeneity. This technique enables the detection and analysis of protein expression at the single-cell level. It provides a more precise view of protein expression compared to its RNA-sequencing counterpart, revealing insights into protein localization and intra-cellular interactions. For instance, immunohistochemistry provides a more direct appraisal of a patient’s tumor condition. This technique plays a pivotal role in breast cancer diagnosis, as it involves staining clinical samples of breast cancer tissues to reveal the expression of crucial proteins, including ER, PR, and HER2. The resultant immuno-stained samples offer a visual map for clinicians, aiding them in identifying the subtype of breast cancer and enabling effective pathological staging and treatment selection. Through techniques like immunohistochemistry, we can not only discern between these subtypes, but also detect signs of lymph node metastasis and monitor potential tumor recurrence. Such insights are pivotal for guiding treatment decisions and tracking the progression of the disease [14,15,16].

In conclusion, breast cancer, as a pervasive global health challenge, necessitates a comprehensive understanding of its heterogeneity and progression. The profound relevance of single-cell detection techniques, particularly the transition from single-cell genomics to single-cell proteomics, opens a new frontier in the exploration of this heterogeneity, thereby paving the way for more precise treatment strategies and reliable prognosis assessments.

## 2. Genomic Profiling

### 2.1. Traditional Genomic Profiling

In 1994, the BRCA1 gene was identified through positional cloning, followed by the discovery of the BRCA2 gene in 1995. These genes play a crucial role in DNA damage repair and maintaining genomic stability, thereby reducing the risk of tumor development [17,18]. BRCA1 and BRCA2 are tumor suppressor genes involved in repairing dsDNA breaks. Mutations in these genes significantly increase the lifetime risk of developing breast cancer. Inherited mutations in BRCA1 and BRCA2 account for a small percentage of breast cancer cases. Tumors associated with BRCA1 mutations often exhibit a basal-like phenotype and a higher histological grade, while those linked to BRCA2 mutations resemble sporadic tumors more closely. Several specific scenarios can notably elevate the incidence of breast cancer: (1) sequence variants encoding premature termination codons such as nonsense or frameshift mutations occurring prior to the 1855th amino acid in BRCA1 and the 3309th amino acid in BRCA2; (2) mutations located at splice site consensus sequences—either the first or second base positioned upstream or downstream of an exon; (3) copy number loss mutations leading to frameshift mutations prior to the 1855th amino acid of BRCA1 and the 3309th amino acid of BRCA2, or mutations eliminating one or more exons not predicted or confirmed to produce functional in-frame RNA isoforms capable of restoring BRCA1/2 gene function; and (4) copy number repeat variations of any size resulting in the duplication of one or more exons, and proven to cause frameshift mutations before the 1855th amino acid of BRCA1 and the 3309th amino acid of BRCA2 [17,19].

Individuals harboring potential pathogenic variants in the BRCA1/2 genes can benefit significantly from timely education and early screening. According to the European Society for Medical Oncology (ESMO) guidelines [20], females identified with having mutations in BRCA1, BRCA2, or other high-penetrance genes should initiate breast cancer prevention education from the age of 18, maintain vigilant awareness of breast conditions, and comply with regular medical check-ups. Physicians advocate for annual clinical breast examinations complemented by breast X-ray imaging and magnetic resonance imaging (MRI) assessments starting from the age of 25 [21]. Despite the insights gained from early genetic knowledge, they did not immediately translate into clinical treatment strategies. The initial clinical data highlighted that BRCA-associated tumors exhibited high sensitivity to poly (ADP-ribose) polymerase (PARP) inhibitors. These inhibitors act on the PARP-mediated DNA damage repair mechanism, thereby disrupting the tumor’s ability to repair its DNA. As of now, PARP inhibitors are primarily accessible through clinical trials [22].

With the deepening understanding of breast cancer and the widespread use of NGS, more relevant genes have been discovered (Table 1).

There was a study involving a large cohort of breast cancer patients who underwent analysis for gene mutations and copy number variations, further substantiating the prevalence of gene alterations in this population [30]. TP53 gene mutations are commonly found in the basal-like subtype, while the HER2-positive subtype also exhibits a high incidence of TP53 gene mutations. Additionally, the HER2-positive subtype shows a significant frequency of PIK3CA gene mutations. The basal-like and HER2-positive subtypes are characterized by genomic instability and susceptibility to changes in gene copy numbers. Conducting concurrent assessments of DNA copy number and gene mutations in breast cancer cells enables the prediction of the cellular subtype. For example, it is known that the amplification of Kras2 is associated with tumor progression, while insufficient Kras2 copy numbers delays tumor progression [31]. In an investigation involving 16 human basal-like breast tumors, none displayed Kras2 mutations; however, an increased DNA copy number at the Kras2 locus was observed in 9 of the tumors. These observations imply that Kras2 amplification may modulate cell phenotypes or earmark target cell types that are susceptible in basal-like tumors [32].

Furthermore, leveraging insights from patients’ DNA sequencing results proves invaluable in tailoring subsequent treatment strategies [33]. Taking the P53 gene as an example, it plays a fundamental role as a key regulator of cellular processes, participating in controlling cell proliferation and maintaining genomic integrity and stability. Activated in response to an array of stress signals, the TP53 tumor suppressor protein curbs cell transformation by precipitating cell cycle arrest, DNA repair, and apoptosis. In breast cancer patients, however, TP53 gene mutations may precipitate a partial or complete functional loss of the TP53 tumor suppressor protein, thereby undermining its capacity to inhibit tumor development. Significantly, Asian breast cancer patients exhibit a mutation frequency of 42.9% in the P53 gene, which surpasses the mutation rate of 30% observed in Western breast cancer. This suggests a potentially higher degree of endocrine therapy resistance and lower survival rates among this demographic. Hence, breast cancer patients with inactivating mutations in the P53 gene necessitate swift intervention with appropriate follow-ups, reexaminations, and immediate treatment. For those carrying TP53 mutations, related treatment strategies can be explored in a clinical setting, which might include Gendicine therapy either as a standalone treatment or in conjunction with radiation therapy, chemotherapy, or hyperthermia, among other treatment approaches. Overall, the use of such targeted treatment for TP53 mutations has achieved a complete response rate of 30–40% and a partial response rate of 50–60% in various clinical applications and studies, with an overall response rate reaching 90–96% [34].

### 2.2. Single-Cell Genomic Profiling

While conventional genetic profiling can offer valuable insights, it faces significant challenges, most notably its inability to differentiate between normal and tumorous tissues. Breast tumors typically comprise a heterogeneous mix of cancerous cells, healthy tissues, stromal components, and infiltrating leukocytes. Histopathological assessments have revealed that certain samples may contain a composition of around 60% normal cells and 35% cancer cells, with a significant presence of infiltrating leukocytes [35]. The information from these additional normal tissues can be considered in the results, which may potentially overshadow crucial information and even lead to erroneous results. Additionally, traditional DNA analysis can only provide vague insights into cancer development because large-scale analyses yield average DNA profiles for tumors, making it impossible to differentiate and track each tumor cell lineage within a tumor tissue [36,37]. Such issues can be addressed through single-cell DNA sequencing [7].

A pivotal step in scDNA-seq involves extracting minuscule quantities of DNA from single cells, followed by whole-genome amplification (WGA). To minimize amplification-related errors and biases, PCR-based methods are commonly utilized for copy number variation (CNV) detection due to their ability to provide more uniform coverage. In contrast, for single-nucleotide variation (SNV) detection, MDA-based techniques are favored owing to their use of high-fidelity DNA polymerases that function at room temperature and display a heightened sensitivity to single-base alterations [38,39].

CNVs are common in a wide range of cancer cell lines, with conventional genomic analyses indicating their substantial influence in the emergence and progression of breast cancer. An assessment and identification of genomic regions bearing copy number alterations—both gains and losses—in tumor cells can facilitate the construction of lineage trees, elucidating shared ancestry among tumor subpopulations [40]. Single-cell DNA sequencing enables researchers to scrutinize CNVs and discern tumor cell subpopulations within solid tumors, even at the early stages of breast cancer development [35]. By employing a comparative analysis of CNV differences, researchers have categorized roughly 100 tumor cells into three distinct subpopulations: a diploid (D) cell subpopulation characterized by a flat morphology, a pseudodiploid (P) cell subpopulation showcasing varying degrees of deviation from diploidy, and a subpopulation embodying complex genomic rearrangements, mirroring the characteristics of a ‘late-stage’ tumor subpopulation. During the late stages of breast cancer progression, liver metastasis invariably becomes a significant concern. By employing pseudodiploid cells as reference standards, researchers have identified striking similarities between the copy number profiles of primary tumors and their metastatic counterparts. This strongly suggests that the origin of metastatic cells can be traced predominantly to late-stage amplifications, as opposed to intermediate stages or entirely divergent subpopulations [35].

Conceptually, information pertaining to SNVs within breast cancer cells can be extrapolated from pre-existing WGA data. While this methodology has proven to be adequate for detecting copy number variations, it falls short in resolving whole-genome mutations at a granular base pair resolution. A commonly used approach is to increase coverage by performing deep sequencing of these libraries. Researchers have developed a high-coverage whole-genome and exome single-cell sequencing method using high-fidelity DNA polymerase to amplify 22 chromosome-specific primer pairs. The amplified DNA is then incubated with Tn5 transposase, which fragments and ligates the DNA for sequencing adapters [41]. This technology achieves a low false-positive rate for point mutations, equivalent to 1–2 errors per million base pairs [42,43]. Using this technique, researchers selected invasive ductal carcinoma from estrogen-receptor-positive (ER1/PR1/HER2) breast cancer patients for bulk and single-cell sequencing. After filtering germline variations, several non-synonymous mutations were identified in the non-diploid tumor cell population, including TBX3, NOTCH2, JAK1, ARAF, NOTCH3, MAP3K4, NTRK1, AFF4, CDH6, SETBP1, AKAP9, MAP2K7, ECM2, and ECM1 [30].

Through a comprehensive analysis of the breast cancer dataset, investigators have pinpointed two key pathways that are notably disrupted during tumor evolution: the TGF-β signaling pathway and the extracellular matrix receptor signaling pathway [7]. This revelation carries substantial clinical implications, as it enables healthcare professionals to select appropriate chemotherapy drugs that specifically target these disrupted pathways. For example, TGF-β receptor I kinase inhibitors like LY2157299 can inhibit TGF-β signaling pathway transmission, slowing down tumor growth and metastasis [44]. Additionally, drugs targeting extracellular matrix receptor signaling are already being used in clinical practice, such as anti-HER2 therapy for HER2-positive breast cancer patients [7]. These findings offer new insights and approaches for breast cancer treatment.

While single-cell gene profiling technology has greatly advanced our understanding of breast cancer tumor cells, offering high precision and efficiency and resolving some standing challenges, ongoing discoveries of genetically and phenotypically unique cellular subpopulations have prompted a shift in our perception of cancer. The cancer narrative has evolved beyond viewing it as a homogeneous aggregation of tumor cells, and we now embrace the reality of it as a spectrum of heterogeneous and evolving cancer subtypes [45]. This underlines that a sole reliance on DNA single-cell detection may fall short in addressing the research demands presented by breast cancer heterogeneity and in guiding clinical interventions directly. Therefore, it is necessary to utilize other single-cell omics detection technologies to collectively address research questions in breast cancer.

## 3. Transcriptional Profiling

### 3.1. Traditional Transcriptional Profiling

While DNA sequencing offers valuable genetic insights into cells, aiding in the identification of cellular subpopulations and potentially predicting tumor development trends, it falls short of fully capturing the functional states and metabolic activities that are pivotal to comprehending cancer cell behavior. Therefore, its correlation with tumor cell behavior is not strong enough and has not become a clinically recognized classification indicator. Only genetically engineered mouse models of breast tumors with the inactivated genes TP53, BRCA1, and RB have been designed to simulate the genetic alterations found in human breast cancer [46]. When contrasted with DNA, transcriptional data provide a more intuitive reflection of the biological processes and functional states within cells. In cancer, different cellular subpopulations often express different genes and metabolic pathways. Therefore, analyzing the transcriptome of cancer can help us better understand the different subtypes of breast cancer and more accurately predict cellular behavior [47,48].

The transcriptome, a comprehensive assembly of RNA molecules expressed by cells at a specific moment, includes messenger RNA (mRNA), functional RNA variants, and stochastic transcripts [49]. Among them, mRNA can be transcribed into proteins and accounts for approximately 1.5% of the total RNA [49,50]. Functional RNA includes various types, among which long non-coding RNA (lncRNA) and small non-coding RNA (sncRNA) are common [50]. lncRNAs, extending beyond 200 nucleotides in length, serve a multitude of functions, encompassing post-transcriptional regulation, chromatin modification, RNA processing and stability, and protein translation, among others [51]. On the other hand, sncRNAs include shorter RNA types, such as microRNAs (miRNAs) and small interfering RNAs (siRNAs), which regulate gene expression and function by modifying other RNA molecules. Through transcriptome profiling, the expression levels of various RNA types can be simultaneously detected, providing insights into gene expression and regulation within cells [52,53].

#### 3.1.1. mRNA Expression Profiling in Breast Cancer Heterogeneity

Gene expression profiling has greatly advanced the categorization of breast tumors. Previous studies using DNA microarrays for transcriptome analysis identified four major intrinsic subtypes of breast cancer (luminal A, luminal B, HER2-enriched, and basal-like) and normal-like breast groups, which exhibited significant differences in incidence, survival rate, and treatment response [54]. However, with the development of gene expression research, breast tumors will be further subdivided into new molecular entities.

In 2007, through the analysis of 13 samples, researchers unearthed a novel molecular subtype dubbed ‘claudin-low’. It is characterized by a low gene expression of tight junction proteins, claudins 3, 4, and 7, as well as the calcium-dependent cell adhesion glycoprotein E-cadherin. Claudin-low tumors respond to standard neoadjuvant chemotherapy at a rate that is intermediate between that of basal-like and luminal tumors [32]. The molecular characterization of the intrinsic claudin-low subtype uncovered developmental structures echoing those of typical epithelial breast tissue. While claudin-low tumors share considerable similarities with basal-like tumors, they display a notable decrease in the messenger RNA (mRNA) expression of the cell cycle gene Ki67, hinting towards a potentially slower tumor proliferation rate. Follow-up studies discerned 1308 and 359 genes that exhibited markedly increased and decreased expressions, respectively, in tumors with a low level of tight junction protein, firmly establishing it as a unique subtype of breast cancer [55].

Previously, normal-like breast cancer, which exhibits a gene profile resembling normal breast epithelial cells, was not recognized as a distinct molecular subtype. Some studies even suggested that it might represent a group of breast cancers that cannot be classified due to potential contamination by normal epithelial cells [56]. However, further investigation isolated cells displaying normal-like breast cell characteristics, which demonstrated insensitivity to certain chemotherapy drugs like the polyamine analogue DENSPM. While DENSPM strongly inhibited the proliferation of breast cancer cell lines due to its impact on polyamine biosynthesis and catabolism, normal-like breast cells swiftly recovered from DENSPM-induced growth suppression without suffering irreversible detrimental effects, suggesting that the isolated cells may indeed represent normal breast epithelial cells [57,58,59].

Further gene expression studies have supported the classification of normal-like breast cells. Utilizing a dataset from 295 women with breast cancer samples, researchers performed an analysis using five gene-expression-based models. Four of the models produced similar predictions, indicating that normal-like breast cells exhibit significant gene expression differences compared to the luminal B subtype and have lower recurrence rates and a relatively favorable prognosis similar to the luminal A subtype [60].

Beyond the uncovering of novel breast cancer subtypes, gene expression profiling has been harnessed to facilitate a more precise and thorough analysis of existing, controversial breast cancer subtypes [61]. Triple-negative breast cancer (TNBC), comprising 10–20% of all breast cancer cases, predominantly presents in younger patients. [62]. Unfortunately, the prognosis for metastatic TNBC remains poor, with less than 30% of patients surviving beyond 5 years, despite adjuvant chemotherapy being the standard treatment. Clearly, there is an urgent need to better understand the molecular basis of TNBC and develop effective therapeutic approaches for this aggressive breast cancer. Traditionally, the majority of TNBCs, ranging from 50% to 90%, were designated as basal-like, which is a classification based on immunohistochemistry (IHC) or associations with intrinsic molecular subtypes of breast cancer [63,64,65]. However, recent studies utilizing gene expression profiling have revealed the heterogeneity of TNBC. In a comprehensive analysis of gene expression profiles from 3247 breast cancer samples, it was discovered that while 47% of TNBCs showed a basal-like gene expression pattern, the remaining 53% were associated with diverse molecular subtypes including luminal A (17%), normal-like (12%), luminal B (6%), HER2 (6%), and some that remained unclassified (12%). This indicates that TNBC is not limited to tumors with a basal-like phenotype, but rather represents a heterogeneous collection of tumors with different phenotypes [66].

Following this, gene expression profiles were used as the basis for distinguishing TNBC into seven distinct subtypes: basal-like 1 (BL1), basal-like 2 (BL2), immunomodulatory (IM), mesenchymal (M), mesenchymal stem-like (MSL), luminal androgen receptor (LAR), and unstable (UNS) subtypes [66]. The BL1 subtype exhibits high activity in components and pathways related to cell cycle and cell division, such as the DNA replication response group. In contrast, the BL2 subtype is involved in growth factor signaling pathways, including the EGF, NGF, MET, Wnt/β-catenin, and IGF1R pathways, as well as in glycolysis and gluconeogenesis processes. The IM subtype is characterized by its primary activation of immune cell and cytokine signaling pathways, including antigen processing and presentation, and core immune signaling transduction pathways. The M subtype is distinctively enriched in pathways related to cellular movement, such as the Rho-regulated actin cytoskeleton, ECM receptor interaction, and pathways associated with cell differentiation, like the Wnt, anaplastic lymphoma kinase (ALK), and TGF-β signaling pathways. The MSL subtype is characterized by distinct patterns of inositol phosphate metabolism, calcium signaling, and unique expression profiles of ABC transporters. Among the TNBC subtypes, the LAR subgroup presents the most differential gene expression profile. Despite being ER-negative, this subtype is notably enriched with genes implicated in hormone regulation pathways, encompassing steroid synthesis, porphyrin metabolism, and androgen/estrogen metabolism. The classification is pivotal, providing critical models for preclinical investigations of emerging targeted therapies. Patients with BL1 and BL2 subtypes could benefit from medications that target highly proliferative tumors, including taxanes and other anti-mitotic and DNA-damaging agents [67]. On the other hand, patients with the MSL/M subtype would demonstrate an enhanced sensitivity to dasatinib [68]. Meanwhile, patients with the LAR subtype would likely respond better to therapies targeting the androgen receptor, such as enzalutamide [66].

#### 3.1.2. mRNA Half-Life Detection in Breast Cancer Heterogeneity

In addition to studying the expression of coding RNAs, there have been a few studies on other properties of coding RNAs, such as the mRNA half-life [12]. Within the cellular environment, gene expression upregulation can stimulate the synthesis of target proteins. However, mRNA molecules that have fulfilled their role are usually degraded by endogenous nucleases, thereby limiting their temporal stability and sustained expression. If post-transcriptional mRNA molecules can reduce their chances of degradation and maintain long-term stability, they can facilitate the synthesis of a large amount of proteins and play an important role in determining cell expression and influencing cell behavior [69]. Several mechanisms can regulate the stability of mRNA, including mRNA-binding proteins and alternative polyadenylation, among others. These mechanisms predominantly act at sites within the 3′ untranslated region (3′UTR) of mRNA, thereby influencing its degradation rate and overall stability [70,71]. The strict regulation of mRNA stability is crucial for cells to execute their normal functions. In tumor cells such as breast cancer cells, changes in mRNA stability often occur, resulting in the excessive production of a series of growth factors, oncogenes, and other mediators involved in cancer progression, thus driving the occurrence and development of tumors [72,73].

The modulation of the mRNA half-life is pivotal in breast cancer [74]. mRNA-binding proteins hasten the degradation of mRNA by binding to its 3′UTR. In its role as a tumor suppressor in breast cancer, the mRNA-binding protein TTP curtails the level of TTP-binding protein, which permits mRNA to persist for an extended duration, consequently hastening the progression of breast cancer tumors. Notably, by reducing mRNA stability, such as by amplifying the expression level of TTP, the risk of death from recurrent breast cancer in patients is slashed by two to three times. Therefore, TTP, which can reduce mRNA stability, is also suggested as a useful negative prognostic indicator for breast cancer [75,76]. Alternative polyadenylation (APA) can also influence breast cancer progression by modulating the length of the 3′UTR, thus augmenting RNA stability. A case in point is the shortening of the 3′UTR, which, as reported by Lembo et al., is linked to poor prognosis [77]. The authors suggest that shorter 3′UTRs may contain fewer regulatory elements that are typically involved in mRNA decay, such as miRNA binding sites or AU-rich elements (AREs). By favoring a shorter 3′UTR, the mRNA isoform’s degradation rate can be lowered by evading potential miRNA- and RBP-mediated decay. This change in the ratio between long and short isoforms can instigate a transition from a normal state to a cancerous state [77,78]. Moreover, hormones, such as estradiol, can also modulate cancer progression by influencing the mRNA half-life. Estradiol, in particular, plays a significant role in the onset and development of breast cancer. A recent study revealed that estrogen also contributes to the regulation of the mRNA half-life. The researchers found that estrogen fosters the event of intracellular APA, which results in the shortening of the 3′ untranslated region (3′UTR) of CDC6 mRNA. CDC6 plays an important role in DNA repair, and by stabilizing CDC6, estrogen ultimately enhances the DNA repair function in tumor cells, promoting the progression of breast cancer [79].

#### 3.1.3. Functional RNAs in Breast Cancer Heterogeneity

In addition to coding RNAs (mRNAs), functional RNAs also hold significant importance in transcriptomic profiling studies of breast cancer. Research has demonstrated the crucial roles of functional RNAs, such as small non-coding RNAs (sncRNAs) and long non-coding RNAs (lncRNAs), in the initiation and progression of breast cancer.

Small non-coding RNAs (sncRNAs) are under 200 nucleotides long and encompass various types of small RNAs. The most common types include miRNAs, piRNAs, rsRNAs, and tsRNAs, each exhibiting different lengths, structures, and biological functions. Though miRNAs were first discovered in the early 1990s, the earnest exploration and research into their role in breast cancer did not commence until 2005, when their presence was detected in breast tumors [80,81]. miRNA genes are widely distributed, with a considerable proportion of them located in specific regions of chromosomes, including vulnerable sites of gene alterations, as well as regions of deletion and amplification in human cancers. This suggests a relationship between miRNAs and genomic changes in breast cancer [82,83]. Recent comprehensive analyses have extensively investigated the aberrantly expressed miRNAs in human cancers. In-depth studies have been conducted on the functions of some of these miRNAs in breast cancer (Figure 3), and specific miRNAs have been identified for predicting and classifying breast cancer subtypes [48,52,84,85,86].

Initially, researchers distinguished normal and transformed cells based on disparities in miRNA expression between breast cancer cells and normal breast cells. In 2005, Iorio and colleagues identified specific miRNAs in breast cancer cells that differed from those in normal cells, enabling a flawless classification of tumors and normal tissues. Their initial study on miRNA dysregulation in breast cancer used a microarray analysis to compare various breast cancer samples with normal breast tissue, identifying 17 upregulated and 12 downregulated miRNAs specific to breast cancer cells. Additionally, miRNA differential expression was observed in ER-positive (11 miRNAs) and PR-positive (7 miRNAs) samples, suggesting the potential of miRNAs as molecular markers for breast cancer subtyping [82]. As research on miRNAs advanced, scientists gained the ability to preliminarily categorize basic types of breast cancer by analyzing the miRNA profiles of tumor cells. In a 2007 study, the differential expression levels of 38 miRNAs in breast cancer tumor samples enabled researchers to classify 51 out of 93 tumor samples, including 16 basal-like tumors, 15 luminal A, 9 luminal B, 5 HER2-positive, and 6 normal-like tumors [87]. Although they could not determine specific miRNAs for each subtype of breast cancer tumor, this finding demonstrates the ability of miRNA features to classify the five molecular subtypes of breast tumors (luminal A, luminal B, basal-like, HER2-positive, and normal-like).

As research has evolved, investigators have shifted their focus from merely distinguishing miRNA expression differences to exploring specific miRNAs associated with various breast cancer subtypes. In 2011, researchers were able to differentiate between HER2-positive patients and TNBC patients. Specifically, through barcode Solexa sequencing, the study investigated 185 breast specimens, including 11 normal breast tissues, 17 ductal carcinomas in situ (DCIS), 151 invasive cancers, and 6 ductal cell lines, and identified miRNAs specific to TNBC. miR-17-92 and miR-19a showed approximately three-fold higher levels in TNBC, while miR-423 and miR-184 levels were significantly lower in HER2-positive patients [80]. Besides the aforementioned breast cancer subtypes and their related miRNA dysregulation, numerous research teams have identified distinct aberrantly expressed miRNAs associated with various breast cancer subtypes. For instance, the expression of miRNA-205 is found to be reduced in the luminal A subtype, whereas miR-206 expression is observed to increase in the luminal B subtype [88]. Furthermore, in the basal-like subtype, miRNA-205 expression is low, while miR-221 and miR-222 expressions are elevated [89,90]. In HER2-overexpressing breast cancer, the expression levels of miR-125a and miR-125b are generally downregulated [91].

Furthermore, with the discovery of an increasing number of miRNAs, these biomolecules have transformed from being merely markers for breast cancer subtyping to being focal points of detailed studies, including investigations into their origin and function. The functions of miRNA families, such as miR-200 and let-7, have been investigated in breast cancer. The miR-200 family is composed of five members, namely miR-200a, miR-200b, miR-200c, miR-141, and miR-429. A recent study exploring the functions of miR-200 identified three miRNA clusters—miR-200c-141, miR-200b-200a-429, and miR-183-96-182—that play key roles in the self-renewal regulation of breast cancer stem cells (CSCs) and the overall functional regulation of stem cells. These miRNA clusters are downregulated in human breast CSCs, normal human cells, and mouse breast cells [92]. The let-7 family, recognized as one of the earliest identified mammalian miRNA families, comprises 12 members: let-7-a1, let-7-a2, let-7-a3, let-7-b, let-7-c, let-7-d, let-7-e, let-7-f1, let-7-f2, let-7-g, let-7-i, and miR-98 [93]. Research has demonstrated that the expression levels of the let-7 family are intimately connected with the behavior of breast cancer cells, showing specific changes in different breast cancer subtypes. This provides a new perspective for analyzing tumor heterogeneity in breast cancer based on miRNA sequencing. In poorly differentiated aggressive breast cancer cells that display a mesenchymal phenotype, let-7 miRNAs are typically absent. Conversely, in more differentiated cells with an epithelial phenotype, the levels of let-7 miRNAs are relatively higher [94]. To be more specific, during different stages of human breast cancer, the let-7 miRNA family is downregulated in DCIS, where cancer cells are confined to the breast ducts without invading the surrounding tissues, and invasive ductal carcinoma, where cancer cells invade beyond the breast ducts into tissues such as the breast parenchyma or lymph nodes. Additionally, a low expression of the let-7 family in ER-positive breast cancer cells leads to the upregulation of ER-α activity, promoting cell proliferation and resisting apoptosis [95].

Interestingly, the analysis of cellular gene expression profiles reveals overlapping areas and zones of ambiguity among the newly identified subtypes of breast cancer, specifically the claudin-low, normal-like, and TNBC subtypes. The basal-like and claudin-low subtypes exhibit certain similarities in gene expression patterns. They both demonstrate low expressions of genes such as HER2, ESR1, and GATA3, as well as luminal and ductal markers like claudins 8 and 18. Additionally, approximately 15–25% of claudin-low tumors show hormone receptor positivity (HR+), and a similar trend is observed in around 10% of basal-like tumors [96,97]. Research proposes a further subdivision of the basal-like subtype into Basal A and Basal B, informed by distinct gene expression patterns in breast cancer. The Basal A subtype predominantly corresponds to the basal-like cells identified in primary tumors, while the Basal B subtype is more diverse, including not only TNBC, but also cells characteristic of normal-like breast cancer [97].

Notably, the investigators observed remarkable similarities between the breast cancer cell lines in Basal B (characterized by a TNBC, normal-like phenotype) and the claudin-low subtype. Both demonstrated low expressions of luminal and HER2 gene clusters as well as cell adhesion clusters that include claudins 3, 4, and 7 and E-cadherin [98]. Moreover, the stem cell marker, aldehyde dehydrogenase 1 (ALDH1), is notably expressed in both normal-like and claudin-low tumors. Also, there is an increased expression of basal cluster markers (KRT6A/B, KRT17, KRT14, and KRT5) in TNBC and claudin-low tumors, exhibiting the defining triple-negative characteristics (ER-/PR-/HER2-) [99]. These findings collectively suggest a high degree of similarity between the claudin-low subtype and the basal-like and normal-like subtypes in TNBC, thereby hinting at the potential existence of distinct or shared novel tumor subtypes.

Further studies have provided additional evidence supporting this viewpoint. In a classification spectrum of 400 TNBC tumor patients, it was found that 72% were classified as basal-like, 9% were classified as HER2-enriched, 6% were classified as luminal B, 5% were classified as luminal A, and 8% were classified as normal breast-like [100]. However, when considering the claudin-low classification, the distribution changed to 49% basal-like, 30% claudin-low, 9% HER2-enriched, 6% luminal B, 5% luminal A, and 1% normal breast-like. This implies that when considering the gene expression profiles of the claudin-low subtype, a portion of basal-like and normal breast-like tumor cells are classified within it. In essence, the claudin-low subtype intersects with the normal breast-like subtype and the basal-like subtype in TNBC, leading to a mixed classification of cancer cells. This ambiguity might arise from the nebulous definition of the basal-like subtype within TNBC, which also encompasses non-basal-like tumor cells. Alternatively, it might suggest a potential new subgroup existing between basal-like and claudin-low tumors, or it could reflect the complexity of the tumor microenvironment, including contamination via normal breast epithelial cells and the infiltration of immune cells [55,68]. Though the implementation of precise microarray gene expression profiling and gene expression clustering models significantly improves the ability to differentiate breast cancer cell subgroups, the intricate heterogeneity of breast cancer remains a complex challenge yet to be fully tackled.

### 3.2. Single-Cell Transcriptome Profiling

Single-cell RNA sequencing (scRNA-seq) requires the prior isolation of individual cells from tumor tissue, in contrast to traditional bulk RNA analysis that involves breaking down large tissue pieces or cell suspensions. Common methods for single-cell isolation include micromanipulation, laser capture microdissection, microfluidic techniques, and fluorescence-activated cell sorting (FACS). Micromanipulation allows for the manual isolation of individual cells while preserving their original states, while laser capture microdissection provides spatial information of target cells [101,102]. Microfluidic techniques, including droplet-based methods, offer high-throughput and cost-effective automation of single-cell isolation, reverse transcription, and pre-amplification, albeit with certain technical limitations. FACS combines flow cytometry with specific fluorescently labeled antibodies to isolate individual cells of interest from heterogeneous cell populations. A careful evaluation of the advantages and disadvantages of these single-cell isolation methods is necessary based on experimental requirements [103,104].

To enhance accuracy and efficiency in scRNA-seq, reverse transcription and amplification stages can utilize techniques such as Poly(A) tailing or template switching for cDNA synthesis, followed by polymerase chain reaction (PCR) or in vitro transcription (IVT) for cDNA amplification [105,106]. PCR offers non-linear amplification after the second strand synthesis, while IVT is a linear amplification method incorporating a T7 promoter into the poly(T) primer [107]. The choice of reverse transcription and amplification strategies depends on experimental needs and preferences.

In selecting an appropriate single-cell isolation method, researchers should consider specific application scenarios and research questions. For studying a large number of samples like undissociated tissue sections, methods such as inDrop [108,109] and Drop-seq [110,111] can be considered, although they may have limitations in transcriptome coverage and detecting lowly expressed genes [112]. For studies involving a small number of cells like circulating tumor cells (CTCs), flow cytometry or micromanipulation followed by manual library preparation in microwell plates can be suitable. These protocols often amplify full-length mRNA using SMART or alternative chemistry, enabling in-depth transcriptome analysis. It is crucial to evaluate the advantages and disadvantages of these methods to ensure the appropriate choice for single-cell sequencing experiments in breast cancer research [113,114,115].

In recent years, single-cell transcriptome analysis techniques have revolutionized the understanding of tumor heterogeneity in breast cancer. These techniques have been extensively utilized to investigate differentiation trajectories, resistance programs, and immune infiltration in tumors [116,117]. Metastasis, a major driver of cancer-related fatalities, underscores the importance of comprehending its initiation and progression to develop effective therapeutic strategies [53]. To identify subpopulations of cells with metastatic characteristics in breast cancer tissue, researchers have turned to single-cell gene expression profiling [53]. They employed FACS combined with the CD298 gene (ATP1B3) as a marker to discern and enumerate human-derived metastatic tumor cells in mouse peripheral tissues [118]. Additionally, multiplex gene expression profiling was performed on individual cells, targeting genes associated with stemness, pluripotency, epithelial-to-mesenchymal transition (EMT), mammary lineage specification, dormancy, cell cycle, and proliferation. The analysis unveiled that early-stage metastatic cells possess significant tumor-initiating capacity and have the ability to generate luminal-like tumor cells, supporting the notion of stem-like metastatic cells giving rise to luminal-like metastatic cells. Moreover, the examination of genes linked to differentiation features in breast cancer tumor single cells revealed differential expression patterns. Early-stage metastatic cells exhibited higher expression levels of basal/stem cell genes (LGR5, BMI1, BCL2, NOTCH4, and JAG1) and lower levels of luminal genes (MUC1, EMP1, and CD24). Prognostic associations were also identified among genes associated with stem-like metastatic cells, indicating their potential for predicting distant metastasis and aiding in prognostic determinations. Insights derived from single-cell analysis can provide valuable guidance for treatment strategies. High levels of MYC and CDK2 were observed in later-stage metastatic cells, prompting the investigation of dinaciclib, a CDK inhibitor known to induce apoptosis in cancer cells with high MYC expression [119]. Remarkably, the treatment with dinaciclib successfully inhibited the progression of metastasis in the majority of mouse tumors after four weeks of administration.

Moreover, in the initial stages of breast cancer metastasis, breast cancer cells undergo a transformation known as EMT. This transformation allows them to enter the circulatory system as CTCs. Characterizing the genotype and phenotype of CTCs may provide a better understanding of tumor evolution and help with the identification of metastatic initiating cells. However, CTCs are exceedingly scarce in the bloodstream, and often present in very limited quantities, sometimes as rare as a handful of cells per milliliter of blood, and studying the transcriptome of CTCs using traditional RNA sequencing techniques is challenging due to their scarcity and the biases associated with low cell numbers [120].

To overcome these challenges, advancements in single-cell RNA sequencing technologies, such as FACS and microfluidic chips, have been instrumental in efficiently isolating and capturing CTCs for analysis [121,122]. Recent studies have successfully applied single-cell sequencing to CTCs derived from patients with metastatic breast cancer. These studies have revealed a high concordance between the gene expression profiles of CTCs and the corresponding metastatic tissue, demonstrating the potential of CTCs as representatives of the metastatic process [123]. In addition, researchers have developed strategies to extract functional CTCs from mice implanted with human tumor xenografts. These CTCs retain their tumor-initiating and metastatic capabilities, providing valuable models for further investigation [124]. An analysis of CTCs from breast cancer patients has shown that the extracted CTCs were predominantly triple-negative breast cancer cells (lacking expression of ER, PR, or HER2). Significant differences in the ER, PR, and/or HER2 statuses were observed between the enriched CTCs and the corresponding primary tumors in primary and metastatic breast cancer patients. The loss of the cell adhesion protein CDH1 in migrating CTCs was consistent with the acquisition of invasive and migratory characteristics, suggesting that the systematic implementation of single-cell CTC analysis could provide new insights into the biology of migrating tumor cells during the process of metastatic dissemination. Additionally, a multi-marker RNA analysis of CTCs at a single-cell resolution revealed distinct CTC subpopulations. An important finding was that CTCs did not cluster based on patients or disease stages (primary cancer vs. metastatic cancer), supporting the concept that these cells belong to subgroups with phenotypes that are fundamentally different from mixed tumor tissues. Studying and phenotyping primary tumors separately may lead to suboptimal treatment choices [45].

In sum, sequencing CTCs offers valuable insights into tumor heterogeneity, the metastatic process, and the evolution of breast cancer. By understanding the transcriptomic profiles of CTCs, researchers can gain deeper insights into the underlying mechanisms of metastasis and potentially identify novel therapeutic targets. Moreover, single-cell RNA sequencing provides higher resolution and accuracy in addressing questions related to breast cancer heterogeneity.

In our previous discussions, we noted that conventional gene expression profiling exposed a considerable overlap between the claudin-low subtype and the normal-like and basal-like subtypes in TNBC. This overlapping area, which we refer to as the Triple Negative Gray Area (TNGA), introduces potential ambiguity in the cellular classifications of these subtypes. The TNGA includes both the claudin-low and normal-like subtypes, as well as the claudin-low and basal-like subtypes in TNBC, along with any shared cell populations.

Recent studies utilizing single-cell RNA sequencing have provided insights into the subdivision of TNBC tumors into six distinct subgroups. These subgroups include (i) a mesenchymal stem cell cluster characterized by a significantly elevated EGFR expression and an increased expression of genes associated with mesenchymal (VIM, ITGB1, and LAMC2) and stem cell-like (ITGA6 and ITGB4) characteristics; (ii) a mitochondrial cluster with an increased expression of genes localized to mitochondria (MRPL genes, MDH2, and TOMM40); (iii) a proliferative cluster exhibiting a high expression of cell cycle genes (MKI67, AURKA/B, and PLK1), with a majority of cells in the G2/M phase of the cell cycle; (iv) an antigen-presenting cluster with an increased expression of genes involved in antigen presentation (HLA-DRB1, CD74, and HLADRA); (v) a basal cluster characterized by an elevated expression of basal cell markers (KRT6A/B, KRT17, KRT14, and KRT5); and (vi) another cluster without unique expression profile features [66,125]. Additionally, an indeterminate component composed of seven clusters is also observed [66]. By employing this classification schema, it becomes possible to segregate tumor cells from TNBC patients into distinct subgroups, effectively highlighting the extensive intra-tumoral heterogeneity. Notably, a compelling correlation is observed between TNBC tumor cells and the basal-like subtype, as evidenced by their shared high expression of the epidermal growth factor receptor (EGFR) and the keratin family members KRT5, KRT17, and KRT14 [8]. Furthermore, there is an evident association between TNBC cells and the claudin-low subgroup. Both claudin-low and TNBC subtypes exhibit the expression of various transcription factors, including FOXC1 and ZEB1, which play crucial roles in initiating the EMT process, enhancing their invasive and metastatic abilities [53].

Moreover, studies probing the tumor microenvironment reveal signs of exhaustion in T cells within the infiltrating immune cell population of TNBC [126,127]. CD8 exhaustion, characterized by immune cell infiltration, has also been observed in the claudin-low tumor subtype [96]. T cells with high exhaustion features are targets for clinical tumor immunotherapy, particularly immune checkpoint blockade targeting PD-1 (PDCD1), indicating sensitivity to PD-1 treatment in these individuals [126,127,128]. There is also a similarity between the claudin-low and normal-like subtypes, which suggests a certain degree of relationship between them. For example, both subtypes show a low expression of luminal and HER2 gene clusters and a high expression of the stem cell marker ALDH1 [96]. Therefore, within the context of TNBC, there exists a distinct subgroup of cells that exhibits shared characteristics from the claudin-low, basal-like, and normal-like subtypes, collectively identified as the Triple Negative Gray Area (TNGA). Remarkably, this particular cell subgroup not only possesses the triple-negative features (ER-/PR-/HER2-), but also demonstrates a low expression of luminal gene clusters, along with distinct stemness and migratory properties.

Building upon the insights gleaned from the aforementioned research, TNGA cells, characterized by their stemness properties, demonstrate shared traits with three distinct tumor subtypes: basal-like, claudin-low, and normal-like subtypes. Consistent with the CSC theory, TNGA cells are believed to occupy an intermediate state. This premise aligns with the established models proposed in previous studies [96]. In model 1, it was observed that most invasive breast tumors can be arranged along the differentiation hierarchy of normal breast ductal epithelium, starting from the claudin-low subtype closest to Mammary Stem Cells (MaSCs), followed by basal-like and HER2-enriched subtypes, and then the two luminal tumor subtypes, with the luminal subtypes closest to mature luminal cells [129]. In model 2, MaSCs give rise to claudin-low tumors, luminal progenitor cells give rise to basal-like tumors, and mature luminal cells give rise to luminal subtypes A and B tumor cells. The speculated origin cells of HER2-enriched tumors could be epithelial cells in a transitional differentiation state, straddling between luminal progenitor cells and mature luminal cells [96]. In model 3, transformed MaSCs, imbued with claudin-low/mesenchymal characteristics, retain capacity for both symmetric and asymmetric division, the latter of which facilitates cell differentiation and halts at certain differentiation stages. The majority of tumors will consist of luminal progenitor/basal-like cells or more differentiated luminal cells, with each molecular subtype having a subset of cells with a low expression of mesenchymal/tight junction proteins [130].

Initially, model 1 encountered challenges primarily due to the observed association between the basal-like and claudin-low subtypes [110]. Subsequently, model 2 encountered skepticism as claudin-low tumors display a minimal expression of luminal genes [96] and stand far removed from luminal A/B, casting doubt on the prospect of direct differentiation into mature luminal subtypes. The ambiguous association between the claudin-low and luminal subtypes may be ascribed to the misclassification of HER2-enriched tumors (bearing similarity to the claudin-low subtype) as luminal subtypes. In specific cases, cancer cells originating from ER and HER2 double-positive tumors undergo ERBB2/HER2 amplification without significant downstream HER2 signaling activation. Instead, these cells primarily exhibit downstream ER signaling activation, resulting in their classification as luminal subtype B. Nonetheless, they retain the potential for downstream HER2 activation and can manifest HER2-enriched characteristics under certain circumstances [8]. However, we support the paradigm of HER2-enriched and luminal differentiation. This is not solely due to the marked differences between the HER2-enriched and luminal subtypes, such as those in the PI3K and NF-kB pathway genes [8], but also within the TNGA context, as luminal subtypes display more extensive differences in gene expression profiles compared to HER2-enriched tumors. For example, their expression of EMT characteristics is much lower than that of TNG cells. Nevertheless, in comparison to luminal subtypes, HER2-enriched tumor cells lie closer to the TNGA continuum and manifest pronounced levels of stemness and recurrence traits [8].

To address these considerations, we propose a revised model that incorporates the normal-like classification and introduces the TNG cell subtype, while highlighting the luminal progenitor cell type as the primary cell type involved in the transformation process (Figure 4). Under this proposed framework, MaSCs in basal-like cancer undergo restricted differentiation, giving rise to luminal progenitor cells. These luminal progenitor cells then undergo further differentiation into ER-enriched progenitor cells, which subsequently evolve into mature luminal cells. ER-enriched and mature luminal cells, exhibiting independent proliferation, contribute to the formation of various cell subtypes. Importantly, luminal progenitor cells also differentiate into the TNGA cell subgroup. Under different inducing factors, such as MET and changes in the immune microenvironment, they further develop into the basal-like, claudin-low, and normal-like subtypes.

Undeniably, additional research is imperative to pinpoint the progenitor cells of each intrinsic subtype. However, single-cell RNA sequencing studies have provided insights into the complexity and heterogeneity of TNBC. The identification of distinct subgroups within TNBC, along with their shared characteristics and relationships, contributes to our understanding of tumor evolution and opens avenues for more targeted and personalized treatment strategies. Continued advancements in single-cell RNA sequencing technology will undoubtedly enhance our knowledge of the breast cancer subtypes and refine existing models, leading to improved diagnostic and therapeutic approaches.

## 4. Protein Profiling

### 4.1. Traditional Protein Profiling

Despite the strides made in the molecular classification of breast cancer through gene expression profiles, its incorporation into routine patient management remains elusive [131]. The primary hurdles are multi-fold. First, successful deployment calls for robust technical and analytical consistency, which is a feat often impeded by the inherent variability of patient samples across disparate laboratories [132,133]. Consequently, clinicians necessitate guidance from a centralized assessment body, such as the American Society of Clinical Oncology (ASCO) and the College of American Pathologists (CAP), to guarantee a standardized approach [134,135]. Second, while there is a plethora of novel biomarkers, their clinical utility is frequently compromised due to their inability to reveal protein concentration, localization, post-translational modifications (PTMs), or their interaction with other proteins [136]. Third, the global accessibility of these biomarkers is inconsistent, with a glaring disconnect with traditional pathological classifications within different healthcare infrastructures. This disparity exacerbates the difficulty in formulating precise recommendations [133,137].

The application of traditional pathological examinations, focused on disease morphology, remains a mainstay in clinical practice for discerning the unique attributes and characteristics of breast cancer [138,139]. To illustrate, during tumor staging and subsequent clinical management, the TNM staging system frequently takes center stage. This system categorizes cancer patients according to three key parameters: tumor (T), node (N), and metastasis (M). The ‘T’ component reflects the size and invasiveness of the primary tumor, ‘N’ denotes the lymph node involvement, and ‘M’ signifies the presence of metastasis. As the TNM stages progress, they indicate a more advanced cancer stage and a prognosis that is less favorable. In detail, ‘T’ encompasses Tx (un-assessable primary tumor), T0 (absence of primary tumor), Tis (carcinoma in situ), and T1-T4 (four stages gauged on tumor size and invasiveness). ‘N’ includes Nx (un-assessable lymph node status), N0 (absence of lymph node metastasis), and N1-N3 (three stages based on extent of lymph node involvement). The ‘M’ component is subdivided into M0 (no distant metastasis) and M1 (presence of distant metastasis) [140]. Following this, the Nottingham Prognostic Index (NPI) plays a crucial role in the treatment and prognostic evaluation. The NPI allocates scores to tumor size, lymph node metastasis, and tumor grade, and subsequently applies weights to these scores to compute an overarching score that predicts the patient’s prognosis. The scores for the tumor size and the lymph node metastasis range from 0 to 3 and from 0 to 2, respectively, while the tumor grade scores span from 1 to 3. The composite of these three scores gives the NPI score, where higher scores are indicative of a poorer prognosis [141].

Significantly, markers identified through IHC are routinely utilized to discern the protein expression levels within tumor cells, thereby playing an integral part in traditional pathological examinations. The essence of IHC lies in the restoration of the three-dimensional structure of proteins, potentially disrupted during the processes of fixation and embedding, facilitated via heat or chemical treatment. This restoration enables antibodies to interact effectively with target proteins. To amplify specificity and sensitivity, certain proteins such as bovine serum albumin or serum are employed to inhibit non-specific antibody binding to non-target proteins in tissues. This step is succeeded by the application of a specific antibody to bind the protein of interest in cellular or tissue samples, eventually leading to a staining reaction through the interaction with a secondary antibody or enzyme [134]. Within the context of breast cancer diagnosis, standard evaluation involves classification according to the expression levels of key receptors such as estrogen receptor α (ER), progesterone receptor (PgR), and human epidermal growth factor receptor 2 (HER2). These categories comprise ER-positive, ER-negative/HER2-positive, and ER-negative/HER2-negative cases (primarily PgR-negative, also known as triple-negative breast cancer) (Figure 5).

Estrogen receptor (ER): The estrogen receptor (ER) is a steroid receptor transcription factor regulated by estrogen in breast tissue [142]. In breast cancer diagnostics, the ER expression status serves as a pivotal biomarker. More than 75% of breast cancer patients present ER-positive tumors, leaving the rest as ER-negative [143]. ER-positive tumors generally manifest with low-grade histology, reduced invasiveness, and improved prognosis, in contrast to ER-negative tumors, which typically exhibit increased invasiveness and a poorer prognosis [144]. The ER expression status also correlates with breast cancer molecular subtypes. ER-positive breast cancers, when compared with HER2-negative and PgR-positive cancers, have molecular profiles closer to normal breast tissue and hence, have a more favorable prognosis [134]. Conversely, HER2-positive and triple-negative breast cancers, which are more prevalent in younger patients, tend to have a higher degree of invasiveness and a poorer prognosis [145]. The presence or absence of ER expression also holds significant implications for the treatment of breast cancer. Endocrine therapies like aromatase inhibitors and tamoxifen are particularly effective in treating ER-positive breast cancer as they inhibit estrogen action, thus curtailing tumor growth. In contrast, ER-negative breast cancers do not respond to such treatments due to their lack of ER targeting [146].

Progesterone receptor (PgR): The progesterone receptor (PgR), regulated by estrogen, is another significant biomarker in breast cancer molecular diagnosis. In ERα-positive breast cancer, approximately half of the patients express PgR [147]. The level of PgR expression plays a vital role in guiding breast cancer treatment and prognosis. Tumors exhibiting low or absent PgR expression typically manifest heightened proliferative and invasive characteristics, alongside an increased risk of recurrence and a poorer prognosis [148]. Conversely, the most common type of breast cancer co-expresses ERα and PgR [147]. However, the predictive ability of PgR is subject to controversy. While some studies propose a predictive role of PgR in treatment response and prognosis, others negate any such association. In scenarios where ERα detection fails, PgR expression, although uncommon, may be classified as a minor subtype due to false-negative ERα or false-positive PgR results. A subset of these cases exhibits low or aberrant PgR expression despite the absence of apparent ERα expression [145,149].

Human epidermal growth factor receptor 2 (HER2): Human epidermal growth factor receptor 2 (hER2) is a transmembrane tyrosine kinase receptor. Its overexpression and amplification in breast cancer are recognized risk factors. HER2-positive breast cancer constitutes approximately 15–20% of all breast cancer instances [150]. This category of breast cancer is typically associated with high invasiveness, rapid growth, and a poor prognosis. The five-year survival rate and overall survival rate for HER2-positive breast cancer patients trail those of HER2-negative patients [145]. HER2 plays a significant role in the onset and progression of breast cancer. The overexpression of the HER2 protein, as a result of HER2 gene amplification, activates a multitude of signaling pathways, including the RAS-MAPK and PI3K-AKT-mTOR pathways. These pathways promote malignant behaviors such as cancer cell proliferation, growth, and invasion [151].

Triple-negative breast cancer: IHC biomarker classification clearly delineates triple-negative breast cancer, identified by the absence of the estrogen receptor (ER), progesterone receptor (PR), and human epidermal growth factor receptor 2 (HER2) proteins [146]. The absence of these protein markers renders the diagnosis of triple-negative breast cancer dependent on exclusion criteria. Furthermore, the unavailability of these markers limits the use of targeted treatment approaches such as estrogen receptor antagonists or HER2-targeted drugs. These constraints reduce treatment options for triple-negative breast cancer, making chemotherapy the primary treatment modality [152].

### 4.2. Single-Cell Proteomic Profiling

Protein biomarker detection is widely accepted in clinical treatment due to the central role of proteins in cellular behavior and their potential for breast cancer classification [6,136]. However, traditional protein biomarker detection faces challenges in accurately distinguishing patients based on clinical pathological criteria, resulting in suboptimal treatments [100]. By enabling protein analysis at the single-cell level, single-cell proteomic analysis offers a viable approach to distinguish and identify rare single cells within breast cancer tumors. This granularity in identification aids in the precise characterization of breast cancer subtypes, thereby informing clinical treatment and prognostic assessment [153]. Pioneering studies in single-cell proteomics leveraged the protein expression levels within breast cancer cells for classification. The methodologies that were predominantly employed were the following:Flow cytometry, which is an approach that quantifies the fluorescence characteristics of individual cells or particles within a fluid stream when exposed to light sources [154]. Cells labeled with fluorescent antibodies are rapidly channeled through a detection region within a flow chamber. Subsequently, these stained cells are stimulated by lasers, and a detector captures the intensity of the emitted fluorescence. Over the decades, since its inception in the late 1960s, flow cytometry has evolved significantly. It has progressed from an initial capacity to measure 1–2 fluorescent substances within cells, to now being capable of analyzing 10–15 fluorescent substances within a single cell, enabling the assessment of entire cellular pathways.Single-cell mass spectrometry (MS), which is a method that offers the potential for a label-free quantitative analysis of the full proteome of a single cell, inclusive of proteins, peptides, and PTMs [155]. One key advantage of MS is that it does not necessitate molecular labeling, and it can attain sensitivity to the femtomolar level for pure proteins. Various mass spectrometry techniques, such as electrospray MS, laser/desorption/ionization MS, and secondary ion MS, are deployed in single-cell research. However, the utilization of MS for single-cell protein analysis faces challenges, primarily due to an inadequate sensitivity to detect the low-abundance proteins that are typically present in single cells.Reverse-phase protein array (RPA), which is a miniaturized protein imprinting technique that facilitates quantitative monitoring of protein expression in hundreds or even thousands of samples concurrently [156]. This method involves archiving whole-cell lysates in a microarray format for detecting proteins of interest via immunological detection. Notably, RPA obviates the need for protein sample separation via electrophoresis, thereby enabling the concurrent analysis of multiple samples. Additionally, RPA requires only a minimal sample volume for multiplex protein detection.

However, these protein detection techniques like flow cytometry, mass spectrometry, and immunoblotting require large quantities of cells for analysis [157]. Other single-cell protein detection technologies, such as capillary electrophoresis, have the limitation of a lower protein capture efficiency, rendering them unable to achieve genuine single-cell sampling and detection [158]. And a variety of microfluidic methodologies have been developed in recent years, contributing to remarkable advancements in single-cell protein expression analysis. These include droplet-based microfluidics [159], microfluidic flow cytometry [160], microengraving [161], and barcoding microchips [162]. Among these techniques, single-cell Western blotting distinguishes itself through its ability to detect protein levels in single cells. It also boasts user-friendliness, cost effectiveness, and the delivery of easily interpreted results. Single-cell Western blotting, which evolved from microfluid Western blotting, is a protein imprinting technique based on microfluidics. It facilitates swift quantitative analysis of protein samples within individual glass microchannels. The procedure incorporates several stages such as sample enrichment, protein sizing, protein fixation (imprinting), and in situ antibody probing. The accuracy and control provided by microfluidic integration enable a superior efficiency and resolution in traditional protein imprinting [163].

In 2014, an enhancement to this technique was introduced by Herr’s team. They employed a 30 μm thick photosensitive polyacrylamide gel for direct single-cell sampling on microscope slides. Consequently, individual cells were settled into microwells, followed by in situ lysis, gel electrophoresis, and photo-induced imprinting for protein fixation. This allowed for the achievement of thousands of simultaneous protein imprints in a short duration [164].

The true potential for classifying breast cancer subtypes based on single-cell protein expression is achieved when proteomic detection techniques can analyze thousands of individual cells within a short period. For instance, in a 2015 study, Herr’s team utilized single-cell imprinting technology to analyze dissociated single cells derived from human HER2-positive breast tumor biopsies. They reported a five- to ten-fold variance in the expression of mTOR, ERK, and eIF4E across 33 tumor cells [165]. The activation of downstream proteins in these subpopulations of breast cancer cells was associated with an increased tumor invasiveness and a resistance to trastuzumab, significantly lowering patient survival rates [166].

In 2020, a technique devised by Herr’s team enhanced the precision and efficiency of single-cell electrophoresis gel detection by transforming the array format into detachable gel blocks. These blocks were then dehydrated and released into an incubation solution [167]. Their approach unveiled a minor cell subtype in a large population of 478 MCF-7 cell lines, expressing the protein ERα46, which was previously overlooked in conventional single-cell detection. Significantly, this breast cancer cell subtype demonstrated resistance to estrogen therapy, thereby providing groundwork for further proteomics-based breast cancer cell subtype classification [168].

Aside from protein expression level analysis, identifying specific protein isoforms can contribute to breast cancer classification and provide insights into disease progression. For instance, ER-α46, an ER isoform with a truncated and activation-deficient C-terminus, has been studied in this context. Typically, adjuvant hormonal therapies, such as tamoxifen (TAM), are utilized to inhibit the overexpression of ER-α66 [169]. However, relying solely on the nuclear overexpression of the full-length ER-α66 as a hormonal therapy indicator may prove inadequate [170]. Post adjuvant hormonal therapy, ER-positive breast cancer heterogeneity led to recurrence in 21% of stage I patients over 20 years, with 14% presenting distant metastases [18]. Given the homology between ER-α46 and ER-α66, isoform-specific antibodies struggle to differentiate between ER-α46 and ER-α66. Nonetheless, their separation can be facilitated through the migration pattern detected in single-cell Western blotting (scWB) [171,172]. In 2021, Herr’s team distinguished between hormone-sensitive and hormone-insensitive breast cancer cell subtypes by examining the expression levels and frequency of single-cell protein isoforms (i.e., ER-α46 and ER-α66). Based on ER-α isoform expression, they categorized traditional MCF cell lines into three subtypes: MCF1 (co-expressing both ER-α isoforms, ~5%), MCF2 (expressing only ER-α66 isoform, ~60%), and MCF3 (expressing solely ER-α46 isoform, ~30%). ER-α46 served as a subtype-specific biomarker responsive to TAM, with MCF2 cells demonstrating the highest sensitivity to E2 and TAM treatment [9]. Additionally, pAKT and pS6 were identified as determinants for differentiating hormone-sensitive MCF-7 from hormone-insensitive MDA-MB-231 subtypes. In triple-negative breast cancer cell lines lacking full-length ER-α66 protein, the majority of cells expressed ER-α46 and exhibited a highly invasive phenotype [173]. A combination therapy involving PIP5K1α/pAKT inhibitors and TAM could potentially increase the hormonal therapy sensitivity in the pAKT/ER-α46 MDA-MB-231 subtype [174].

Furthermore, by isolating and capturing CTCs from patients’ blood samples, the need for traditional tissue biopsies is eliminated. Although genomics has uncovered a considerable amount of information, there is often a weak correlation between genomics/transcriptomics and protein expression in certain cases [175,176]. It is noteworthy that disparities have been observed in the protein expression between primary tumors and CTCs [177], suggesting that detection protein in CTCs and tumors may provide valuable information. Utilizing a microfluidic single-cell protein imprinting analysis of the nuclei of individual CTCs from ER+ breast cancer patients, Sinkala et al. found that a subset of tumor marker-positive cells also expressed CD45+, indicating the presence of up to 35% of tumor-associated leukocyte subpopulations [178]. Additionally, researchers identified two subpopulations within CTCs derived from ER+ patients based on glyceraldehyde 3-phosphate dehydrogenase (GAPDH) levels, and this was confirmed through a statistical analysis [178]. However, the method of individually selecting CTCs using micromanipulators and transferring each cancer cell to individual microwells limits the throughput of the research. To address this issue, Ding et al. developed a novel SAIF (Selective Adhesion and Immunofluorescence) chip that relies on the unique fluidic behavior of CTCs to achieve high-throughput, label-free, and continuous separation of cancer cells (A549, MCF-7, and HeLa) from leukocytes. This provides a simple and efficient solution for the efficient, label-free, and high-throughput isolation of CTCs from blood cells [179].

With the advancement of microfluidic single-cell protein imprinting, Ding et al. proposed several solutions for microfluidic CTC sorting in 2022, including the antibody-functionalized microfluidic chip (AFM) [180], single-cell immunoblotting microfluidic chip (ieSCI) [181], and a sickle-like inertial microfluidic chip (Orcs-proteomics) [182]. These methods minimized the loss of cytoplasmic proteome and markedly enhanced the utilization of rare cells. In addition, an efficient proteomic analysis of the collected CTCs is equally crucial. Johanna et al. conducted an in-depth proteomic profiling of CTCs isolated from blood samples. In their research, clinical blood samples were collected from two advanced-stage breast cancer patients. They identified 1135 protein groups from seven isolated CTCs and 973 protein groups from five CTCs, with the proteins involved in metabolic pathways being dominant, including 11 tumor markers (PTEN, CD44, BCL2, EPCAM, vimentin, Ki-67, cyclin B1, PRB, AKT, IgG, and K14) [183]. By adopting single-cell protein detection technology, they were able to garner more comprehensive protein data from CTCs, thus uncovering cellular heterogeneity. Furthermore, they could even track individual responses to treatment in clinical studies through the proteomic information gleaned from CTCs.

## 5. Conclusions

Breast cancer tumor heterogeneity is a pressing global issue impacting women’s health. To further decipher and differentiate these tumors, researchers have delved deeply into using techniques such as genomics, transcriptomics, and proteomics. These methods facilitate the analysis of genetic information, gene expression, and protein expression levels in tumor cells, forming the basis of breast cancer classification. Salient achievements in this field include the discovery of six molecular subtypes of breast cancer and three key protein signals.

Markers that are intimately linked with cancer cell behavior have proven to be particularly valuable in breast cancer classification, offering more precise subtype information. Studying transcriptomic products and protein expression associated with disease progression and treatment response in breast cancer cells can yield more effective insights than genomics alone. Owing to the stochastic nature of gene expression, cells with an identical genotype and copy number display variations in their RNA and protein content, implying a divergence between proteomics and transcriptomics. This realization underscores the necessity for proteomic analysis. These aid researchers in better grasping the subtype characteristics of breast cancer, thereby guiding personalized treatment and prognosis assessment. Simultaneously, single-cell detection technology has made significant strides in addressing the issue of breast cancer heterogeneity. By leveraging the groundwork of prior research, researchers can now conduct comprehensive analyses of individual tumor cells using single-cell genomic, transcriptomic, and proteomic techniques, unmasking intra-cellular differences and heterogeneity.

Nonetheless, overcoming breast cancer heterogeneity and actualizing truly personalized treatment necessitates an in-depth understanding of all the molecular characteristics of both the breast cancer and the patient, which is a task riddled with challenges. From the perspective of multiplicity in single-cell proteomics analysis, the comprehensive detection of the membrane, intracellular, and secreted proteins is a prevailing trend. This integrative analysis provides a more comprehensive view of protein information, enabling us to delve deeper into the composition and functionality of cells. However, within a single cell, there exists a vast number of proteins (>10,000). Existing multiplexing methods, though, typically target only one or two specific types of proteins, lacking the ability to cover a comprehensive understanding of protein–protein interactions, signaling networks, and regulatory mechanisms within cells. It is only through the comprehensive analysis of proteins from different types and subcellular locations that we can better elucidate the intricate protein interaction networks within cells and their significant roles in cellular function and disease development [16,184]. For instance, Ding et al. made advancements in flight time cytometry (CyTOF), an emerging powerful proteomics analysis technique that utilizes metal-chelating polymers (MCPs) as mass tags to simultaneously interrogate high-dimensional biomarkers across millions of individual cells. They improved this technique by designing a novel metal-labeled aptamer nanoprobe (MAP), which exhibits higher sensitivity compared to antibodies with a lower epitope coverage, making it safer and more convenient to use [185]. Furthermore, they developed a new metal-labeling strategy based on metal–organic frameworks, enabling multi-parametric and sensitive single-cell biomarker studies. This approach holds great promise as the next generation of molecular probes in mass cytometry. These advancements provide the foundational technological capabilities for multiplex protein imaging [186]. Additionally, spatial information plays a pivotal role in characterizing single-cell proteomics by capturing the protein localization, cell phenotype, and dynamics. It offers valuable insights into cellular interactions and tissue organization, enhancing our understanding of disease progression and cellular functions. However, studying solid tumors poses challenges in handling single-cell proteomic samples. Common analytical approaches involve generating single-cell suspensions through proteomic methods. Yet, the digestion and dissociation processes involved in this procedure can result in the loss of spatial information. These processes disrupt intercellular associations and positional cues, obscuring the true expression programs and vital information about cellular interactions. As a consequence, we may overlook crucial interactions and relative positions of cells within the tumor microenvironment, impeding the accurate localization of distinct cellular subpopulations with varying functions and expression programs in solid tumors [112,187]. Therefore, the primary consideration is to preserve cellular spatial information as much as possible. For example, researchers have been combining tissue sectioning with single-cell proteomics techniques to maintain the spatial positioning of cells within the tissue. By analyzing the individual cells in tissue sections, they can uncover the interactions between cells and the influence of the microenvironment [188]. The second consideration is the optimization of tumor dissociation to generate a cell suspension that fully represents the tumor in terms of cell population, frequency, and expression programs. Ding et al. employed a strategy that is different from the commonly used dimensionality reduction approach. They projected the raw data into a sparse high-dimensional space and improved the clustering characterization of existing methods, such as PhenoGraph. These latent variables, which are highly correlated with only a subset of cells, facilitated the differentiation of cell populations while ensuring the optimal utilization of label information, revealing novel heterogeneity hidden within cell clusters [189].

Moreover, it is essential to meticulously scrutinize multiple ‘omics’ strata associated with the patient and their malignancy, inclusive of genomics, transcriptomics, and metabolomics. Such integrative ‘omics’ approaches have had a far-reaching impact in the domain of cancer research. For instance, as early as 2012, by investigating a plethora of lung cancer patient samples, Peifer and colleagues discovered that the genes CREBBP, EP300, and MLL, which are involved in the histone modification pathway, were mutated, thereby adversely affecting histone integrity, primarily in small-cell lung cancer (SCLC) with a poor prognosis. Hence, besides the TP53 and RB1 gene mutations, their studies hinted that histone modification is another critical characteristic of SCLC. This implies that the concurrent exploration of genomic and proteomic changes can corroborate each other in discerning SCLC [190]. Subsequently, with the advent of single-cell sequencing technology, researchers were able to generate high-resolution tumor microenvironment maps, characterizing tumor cell heterogeneity and establishing tumor evolution histories, by combining traditional bulk transcriptomics with single-cell genomics. Moreover, conducting a prospective analysis of patient tumor samples, by assessing integrative ‘omics’ and functional readouts, can provide evidence to support clinical decision making, thereby augmenting patient survival rates [191]. As single-cell technologies matured, multi-omics cancer detection at the single-cell level has emerged. Nevedomskaya and colleagues utilized various ‘omics’-based platforms, including genomics, epigenomics, cistromics, transcriptomics, proteomics, and metabolomics, to meticulously scrutinize prostate cancer etiology and progression, distinguish benign prostatic hyperplasia from malignant prostate cancer, and ascertain whether local lesions would metastasize distally. Simultaneously, they confirmed that gene copy numbers, DNA methylation, or transcript abundance cannot reliably predict the proteomic changes that occur during prostate cancer progression, highlighting the necessity of multi-omics [192].

These multi-omics studies in other cancer domains have also illuminated the path for investigating breast cancer heterogeneity. Initially, researchers integrated traditional bulk technologies with single-cell technologies to conduct a multi-omics analysis of breast cancer heterogeneity. For instance, Yu et al. employed traditional bulk techniques to obtain relevant information on breast cancer metabolomics. Then, by integrating single-cell transcriptomic information, they analyzed the energy-related metabolic features in breast cancer, successfully stratifying breast tumors into two prognostic clusters: Cluster 1 displays a malignant, high glycolysis activity, and Cluster 2 is characterized by benign tumors that are rich in fatty acid oxidation [193]. Furthermore, with the maturation of single-cell sequencing technology, the use of multi-omics analysis entirely at the single-cell level has facilitated research on breast cancer heterogeneity. There is growing evidence to suggest that the integration of multiple genetic datasets is vital for accurately decoding biological information [194,195]. For instance, Elisabet and colleagues successfully identified nucleic-acid-level events (like SNPs, alternative splicing, or post-translational protein modifications) alongside proteomic profiling in CTCs by marrying single-cell proteomics and single-cell genomics, a method with potential applications in early detection, diagnosis, and treatment [196]. Even further, with the recent advances in machine learning, high-throughput, high-dimension multi-omics research can now be more effectively applied to studies on breast cancer heterogeneity. Stephen and his team developed a multi-omics machine learning predictor for breast cancer treatment response by associating genomic and transcriptomic features with clinical treatment, revealing that malignant cell characteristics, immune activation, and evasion features are related to treatment response. The predictive model’s accuracy was validated in independent external cohorts, suggesting that such multi-omics machine learning methods can be extrapolated to other cancer research contexts [197].

As we forge ahead, our increasingly comprehensive understanding of the molecular characteristics of breast cancer paves the way for the gradual implementation of personalized medicine through prospective trials (Table 2). This will allow for the provision of more individualized treatment plans for patient subgroups and address the challenge of breast cancer heterogeneity.

## Figures and Tables

**Figure 1 cancers-15-04164-f001:**
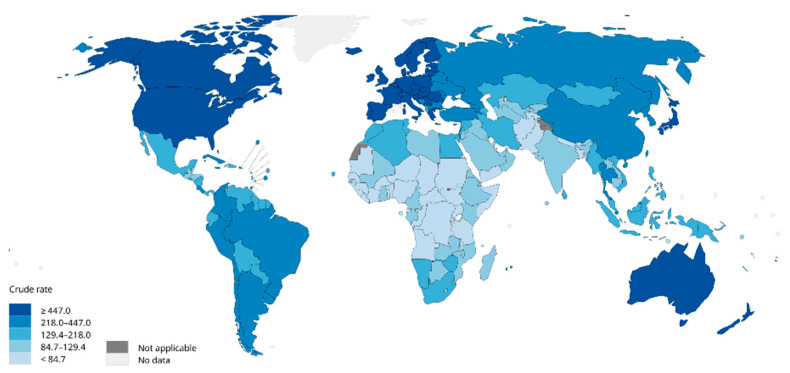
Worldwide estimated crude incidence rates of female breast cancer (2020) [2].

**Figure 2 cancers-15-04164-f002:**
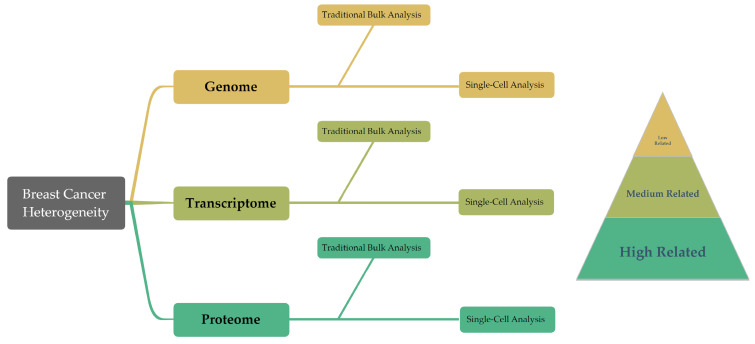
This diagram presents an overview of the structure of the article, summarizing the traditional bulk techniques and single-cell findings in genomics, transcriptomics, and proteomics. Moreover, these three groups demonstrate a progressive correlation with cancer cell behavior in the research on breast cancer heterogeneity.

**Figure 3 cancers-15-04164-f003:**
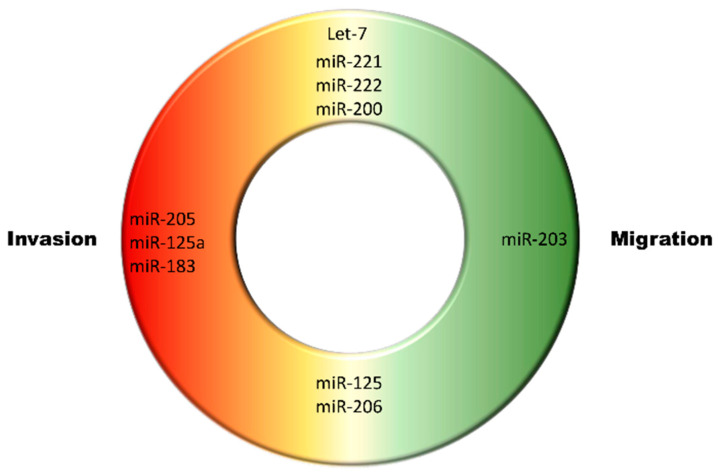
Continuous rings showing an overlapping role of several miRNAs in breast cancer migration and invasion [85].

**Figure 4 cancers-15-04164-f004:**
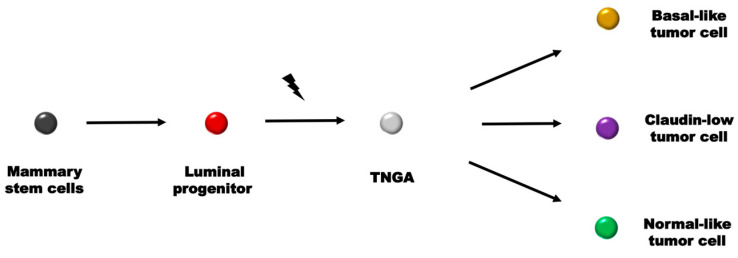
Possible developmental origins of the intrinsic breast cancer subtypes [96].

**Figure 5 cancers-15-04164-f005:**
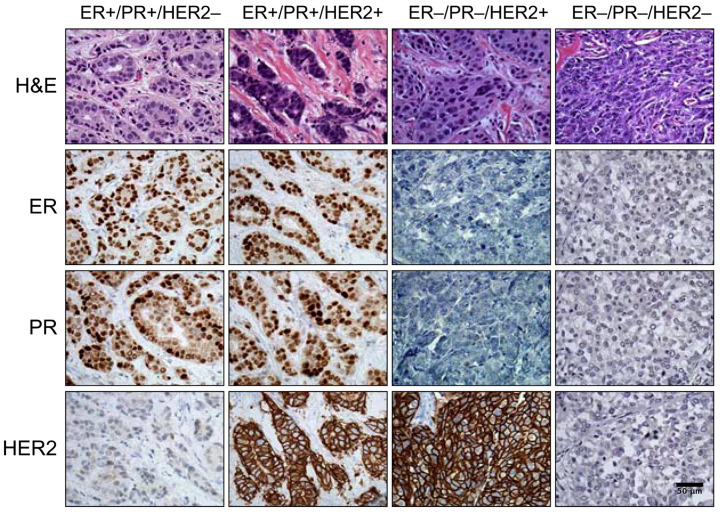
Clinical classification of invasive breast cancer based on expressions of ER, PR, and HER2 [6] (reprinted with permission from Elsevier Ltd.).

**Table 1 cancers-15-04164-t001:** Breast-cancer-relevant gene, discovery year, involved process, and mutation risk.

Gene	Discovery	Involved Process	Mutation Risk	Reference
PTEN	1997	apoptosis, cell cycle, and signal transduction	activation of proliferation and survival signals	[23]
STK11	1997	cell cycle, metabolism, and energy balance	activation of cell proliferation and metabolic pathways	[24]
CHEK2	1999	DNA repair and cell apoptosis	impairments in DNA repair and cell apoptosis processes	[25]
PIK3CA	2004	regulation of signaling pathway	activation of survival signals	[26]
AKT1	2007	regulation of signaling pathway	activation of cell proliferation and survival signals	[27]
BARD1	2010	DNA repair and cell apoptosis	increased susceptibility to breast cancer	[28]
NF1	2015	regulation of signaling pathway	increased rate of developing breast cancer	[29]

**Table 2 cancers-15-04164-t002:** Comparative evaluation of single-cell and bulk ‘Omics’ approaches.This table systematically details the advantages and limitations of single-cell and bulk analysis methods across three major ‘omics’ disciplines: genomics, transcriptomics, and proteomics. The comparison highlights the unique strengths and challenges of each method, with particular emphasis on their ability to capture cellular heterogeneity, the depth and breadth of the molecular information provided, and technical considerations such as cost, data complexity, and analytical robustness in the context of biomedical research.

Omics Field	Analysis Type	Advantages	Limitations	Refs.
Genomics	Bulk	Lower cost; matured analytical methods; provides comprehensive sequence information.	Averages over cell populations; misses information about rare cell populations; limited prediction of the ultimate biological effect.	[19,22,23,24,25]
	Single cell	Detects mutations and structural variations in individual cells; highlights cell-to-cell heterogeneity and rare cell populations; enables study of intra-tumoral heterogeneity in cancer.	Requires substantial sequencing depth for accurate results; higher costs; greater complexity of data analysis; limited information on the ultimate biological effect.	[7,43,44,45]
Transcriptomics	Bulk	Lower cost; matured techniques and analytical methods; global expression analysis; detects all splice variants.	Averages over cell populations; misses cell-to-cell heterogeneity; only represents an intermediate step; correlation with protein levels is not always linear.	[32,57,58,59,82,83,84]
	Single cell	Captures cell-to-cell variability in gene expression; detects all splice variants; sensitive, high dynamic range, and quantitative; parses cell-specific transcriptomes in single-cell experiments.	Data can be noisy; more complex data analysis; only represents an intermediate step; correlation with protein levels is not always linear.	[107,110,116,117,118]
Proteomics	Bulk	Comprehensive coverage of the proteome; mature techniques; resolves the final regulatory level.	Averages over cell populations; less sensitivity to low-abundance proteins; certain proteins difficult to isolate; high dynamic range of proteome makes detection difficult.	[137,141,145,147]
	Single cell	Potential to capture protein-level heterogeneity across individual cells; proteins are the main effectors of cellular function.	Technically challenging; limited coverage of the proteome; less mature techniques; certain proteins difficult to isolate; high dynamic range of proteome makes detection difficult; post-translational modifications may greatly influence activity but can be challenging to analyze.	[160,161,162,163,170,175,180]

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
