# Peer review of "Advancing Breast Cancer Heterogeneity Analysis: Insights from Genomics, Transcriptomics and Proteomics at Bulk and Single-Cell Levels"

_cancers, 2023, doi:10.3390/cancers15164164_

Round 1
Reviewer 1 Report
I like the idea of this article but I have some issues with the execution. Unfortunately, I think this article needs to be reorganized a great deal. I understand that these comments are broad but I will list them below.
The title of this review is misleading. Only about a quarter of the text is about single cell technologies.
The review reads like a dissertation and not a journal article. If the main point of the article is to summarize the use of single cell omics profiling in breast cancer a significant portion of the manuscript is used on summarizing breast cancer broadly and on non-single cell resolution technologies.
This review article is difficult to navigate. I would suggest adding a figure 1 with a roadmap showing the logical progression of each section.
The figures are too simple. The authors cover a lot of information that will be difficult to digest as a reader if the information is not also shown as a figure. For instance, the authors talk a great deal about the different molecular subtypes of breast cancer but there is not a figure clearly showing the receptor statuses etc. of the different subtypes. A second example is that there are many molecular techniques used for single cell analysis, there should be a figure showing how these techniques work. What is the difference between FACS, smartseq2, 10x genomics, mars-seq? These could all be easily shown in a figure.
There is a lot of missing information. The authors need to describe all of the single cell technologies used on breast cancer. Some but by no means all would include: smartseq2, 10x Genomics Chromium, 10x Genomics Multiome, Nanostring CosMx, 10x Genomics Xenium, and seqFISH.
Besides summarizing the techniques, the major discoveries that were only possible because of single cell technologies also need to be enumerated. There needs to be at least a few figures showing the single-cell resolution cellular subtypes discovered with single cell technologies.
Reviewer 2 Report
cancers-2458660
Single-Cell Omics Profiling in Breast Cancer Heterogeneity
The manuscript by Zhu et al. summarized genomics, transcriptomics, and proteomics findings in breast cancer studies. It also highlighted the importance of single-cell multi-omics for future advancements in the field of breast cancer research and management. Overall, the manuscript is comprehensive and well-organized. The authors presented both traditional and single-cell genomics, transcriptomics, and proteomics findings in breast cancer research in recent years. However, this manuscript can be further improved. The authors may consider the following comments to revise the manuscript.
1. The authors should summarize the traditional and single-cell genomics, transcriptomics, and proteomics findings in breast cancer research into tables to summarize and highlight the primary outcomes of the previous studies.
2. Are there any multi-omics studies in breast cancer research?
3. The authors should include a section highlighting the importance of single-cell multi-omics. Here, findings of some typical studies (not necessarily in breast cancer) can be discussed to illustrate the power and potential of single-cell multi-omics.
4. Figures 1, 2, 3, and 4: figure legends contain citations. If the figures were adapted elsewhere, the authors should include a statement of reprint permission and copyright.
5. Table 1: The authors should cite the reference(s) for each row.
6. Lines 45, 163, and some other places: citation error.
Reviewer 3 Report
The review summarizes the analysis and results obtained using various platforms in Breast cancer research. I think the review is timely and will be useful to the readers. I think Figure 2 is not necessary. Rather a summary of what is presented in the review diagrammatically will be appreciated. There are references in the manuscript which shows error. Please rectify it
English language is fine.
Reviewer 4 Report
The article is quite interesting. I suggest the following changes.
* Error in the lines: 45, 163, 290, 444, 733, 803.
*HER2E appears in lines 168, 169 and 170. What subtype is it? It does not correspond to what has been explained above. The same happens with HER22 which appears on line 248.
* Kras2 and KRAS2? are the same. Lines 173 and 177.
*On line 98, the abbreviations of the different RNAs are given and it is indicated what these abbreviations correspond to in section 3.1. This should be corrected.
*Rewrite the paragraph between lines 976 and 990, as not all the research is by the same author. Different works by different researchers are mixed up.
*In lines 971, 976, 1026 and 1055 it is written Ding's team, the correct spelling would be Ding et al. It should be checked beforehand that the research corresponds to this author.
*References 29 and 147 in the bibliography are not correct.
*A summary table could be provided for a better understanding of the work. Sometimes it is difficult to understand each section, because of all the data provided.
Moderate editing of English language required
Round 2
Reviewer 2 Report
The manuscript was appropriately revised and can be accepted.
Reviewer 4 Report
The revised version of the article has improved the quality of the article, making it easier to read and understand.